# Computational Mutagenesis of GPx7 and GPx8: Structural and Stability Insights into Rare Genetic and Somatic Missense Mutations and Their Implications for Cancer Development

**DOI:** 10.3390/cancers17010105

**Published:** 2024-12-31

**Authors:** Adebiyi Sobitan, Nosimot Buhari, Zainab Youssri, Fayuan Wen, Dawit Kidane, Shaolei Teng

**Affiliations:** 1Department of Biology, Howard University, Washington, DC 20059, USA; 2Department of Physiology and Biophysics, Howard University College of Medicine, Washington, DC 20059, USA

**Keywords:** glutathione peroxidases, somatic mutations, cancer, saturated computational mutagenesis, protein stability

## Abstract

This study explores the structural and stability impacts of rare genetic and somatic mutations in two glutathione peroxidase proteins, GPx7 and GPx8, which play crucial roles in cellular stress responses. This study aims to identify how these mutations affect protein stability, as these changes are often associated with cancers and other complex diseases. Using advanced computational methods, this research analyzed thousands of potential mutations to predict their effects on protein function. The findings reveal that destabilizing mutations are more likely to be associated with diseases, providing insights into the molecular mechanisms of cancer development. By focusing on high-impact mutations, this study provides a foundation for developing targeted therapies and better understanding the molecular basis of diseases associated with these proteins.

## 1. Introduction

Glutathione peroxidases (GPxs) are cytosolic enzymes whose function is to catalyze the reduction of hydrogen peroxide to water and oxygen. There are currently eight known mammalian GPxs, most of which are selenoproteins that often simultaneously occupy different cell types [1]. However, certain variants of the GPx family, such as GPx-7, and GPx-8, lack the selenocysteine residue and have instead replaced it with a cysteine residue. The swapping of selenium in selenocysteine for sulfur in cysteine is the differing element between the two amino acids [2].

Both GPx7 and GPx8 are present in the endoplasmic reticulum and play a part in the oxidative folding of endoplasmic reticulum proteins. GPx7 modifies cysteine residues and functions as a stress transmitter, passing along signals to its associated proteins, such as GRP78 and PDI, via disulfide bond transfer in response to various stressors. Some diseases associated with GPx7 are testicular torsion and Keshan disease, both of which are associated with GPx8 as well [3]. A mutation in GPx7 could possibly prove detrimental to its function regarding the cellular processes of apoptosis and tumor suppression. Its enzymatic function and interaction with its associated proteins could result in the dysregulation of oxidative stress responses and in turn promote disease processes. Furthermore, the aberrant function of GPx7 has been linked to ROS accumulations, highly elevated cancer incidences, auto-immune disorders, and obesity [4]. In a pan-cancer analysis of multiple cancer, GPx7 was overexpressed and posed as a potential biomarker for glioma prognosis [5]. However, GPx7 is thought to act as a tumor suppressor in non-malignant esophageal cells [4]. In contrast, GPx8 is overexpressed in gastric cancer and non-small lung cancer cells (NSLCCs). GPx8 is also part of the GPX8/IL-6/STAT3 axis that helps to suppress the progression of an aggressive type of breast cancer [6].

The type II transmembrane proteins, GPx8, possesses a high sequence similarity with GPx7, one of which is their KDEL-like endoplasmic reticulum retrieval motif. They both also exhibit low GPx activity due to their lack of a GSH-binding domain. The physiological function of GPX8 is still unclear; however, it has been reported to be involved in diverse physiological processes [6]. It is worthy to note that GPx8 plays a role in the regulation of calcium (Ca^2+^) in the endoplasmic reticulum. Similarly to GPx7, GPx8 mutations can alter the enzymatic function and its interaction with its associated proteins, possibly leading to the dysregulation of oxidative stress responses, but also endoplasmic reticulum stress responses. However, just like with GPx7, or any mutation, to predict its effects more accurately, you would need to know the type of mutation, where it is in the gene, and how it would affect protein function.

The impact of rare mutations on the proteins GPx7 and GPx8 has not been extensively studied. The primary objective of this study is to employ computational methods to investigate the impact of missense mutations on the structural integrity and stability of GPx7 and GPx8 proteins. Additionally, we aim to assess the carcinogenic somatic mutations in GPx7 and GPx8, with a specific focus on predicting their influence on protein stability. Computational methods play a crucial role in predicting the potential structural and functional consequences of missense mutations, particularly in terms of their ability to disrupt critical active sites involved in protein–protein interactions. These interactions are essential for the enzymatic functions of glutathione peroxidases. By employing free energy calculations and stability prediction algorithms, computational tools enable a more accurate estimation of the risk to protein stability and the destabilizing effects that missense mutations may induce. We can identify and characterize potentially pathogenic mutations that could lead to disease utilizing computational methods. This analysis aids in enhancing our understanding of genetic disorders at the molecular level. Such insights into the molecular basis of diseases can contribute to improved clinical diagnosis and treatment strategies. In particular, the study’s findings provide a valuable framework for prioritizing mutations with significant stability impacts for further functional validation, which could be pivotal in refining diagnostic markers and therapeutic targets in clinical settings. Furthermore, combining the computational results with the experimental data derived from structural studies or biochemical assays can strengthen and validate the findings generated by each approach. This integration of computational and experimental approaches provides a more comprehensive understanding of the detrimental effects that missense mutations may have. In conclusion, this study highlights the importance of computational methods in assessing the impact of missense mutations on protein structure and stability. The findings obtained through computational analysis, when combined with the experimental data, contribute to a deeper understanding of the potential implications of missense mutations in disease development and progression.

## 2. Materials and Methods

### 2.1. Selection and Alignment of the Sequence and Structure of GPx7 and GPx8

We searched the UniprotKB database and filtered for “human” and “reviewed” sequences. The Uniprot IDs for GPx7 and GPx8 were Q96SL4 and Q8TED1, respectively. We selected the canonical FASTA sequences for GPx7 and GPx8 and used the Clustal Omega tool for pairwise alignment. The UniprotKB provided access to available structures for GPx7 and GPx8. We chose a crystal structure determined by X-ray diffraction for GPx7 (PDB ID:2P31), and a predicted AlphaFold structure was selected for GPx8 (AF-Q8TED1-F1). We used PyMOL (version 2.5.4), a molecular visualization tool, developed by Schrödinger for structural alignment.

### 2.2. Phylogenetic Analysis of the Glutathione Family of Proteins

We used the Clustal Omega [7] multiple sequence alignment to construct the identity matrix and determine evolutionary relationships between GPx1, GPx2, GPx3, GPx4, GPx5, GPx6, GPx7, and GPx8. We used the canonical sequences of all the glutathione proteins in humans.

### 2.3. Collection of Cancer-Causing Somatic Mutations

We curated the somatic mutations of GPx7 and GPx8 from the COSMIC database (v99) developed by the Sanger Institute [8]. We searched by gene name and filtered for only missense mutations. We also conducted a curation and analysis of missense mutations and their associated Mendelian traits for GPx7 and GPx8. We collected the genetic mutations from the Human Gene Mutation Database (HGMD) [9].

### 2.4. Saturated Computational Mutagenesis

To generate the number of possible missense mutations that can occur in a protein sequence, we used a custom python script to mutate each residue in GPx7 and GPx8 to 19 other residues. Our approach was structure-based, so we needed the FASTA sequence of the protein structures. The FASTA sequence of the GPx7 structure has (Q23-R177) 154 residues; therefore, the total number of missense mutations generated was 2926, which can be calculated as 154*19. GPx8’s structure has 209 residues, and we generated 3971 (209*19) missense mutations.

### 2.5. Calculations of Gibbs Free Energy Changes

Gibbs free energy during protein folding contributes to the stability of the folded conformation. It is important to note that the stability of a protein is determined by the difference in the Gibbs free energy (ΔG) between folded and unfolded states (ΔG = G^unfolded^ − G^folded^). Foldx (version 5.0) generates numerous structures and computes the Gibbs free energy of the mutant structures and their corresponding wild-type structures [10]. The first step is to repair the raw structure by reducing the conformational energy of each residue using the RepairPDB command (https://foldxsuite.crg.eu/command/RepairPDB (accessed on 3 June 2024)). The syntax for running RepairPDB from the command terminal is:

FoldX --command=RepairPDB --pdb=2P31.pdb

The second step is to run the BuildModel command to compute the energy changes caused by each mutation (https://foldxsuite.crg.eu/command/BuildModel (accessed on 3 June 2024)). The syntax for running the BuilModel command is:

FoldX --command=BuildModel --pdb=2P31_Repair.pdb –mutant-file=individual_list.txt

Ultimately, Foldx outputs the energy changes (ΔΔG) between the Gibbs free energy of each wild-type and mutant protein structure, ΔΔG = ΔG^Mutant^ − ΔG^Wildtype^. A more negative ΔΔG indicates a more stable protein, while a less negative or positive ΔΔG suggests a less stable or destabilized protein. Compared to the experimental ΔΔG values, Foldx predicted ΔΔG values deviated by 0.46 kcal/mol (~0.5 kcal/mol). As a result, a missense mutation is classified as highly destabilizing (ΔΔG > 2.5), destabilizing (2.5 > ΔΔG > 0.5), neutral (0.5 > ΔΔG > −0.5), stabilizing (−0.5 > ΔΔG > −2.5), or highly stabilizing (ΔΔG < −2.5).

### 2.6. Comparative and Statistical Analyses

In our study, we conducted a comparative bioinformatic analysis of all the computed missense mutations of GPx7 and GPx8 utilizing two tools: Meta-SNP and Alphamissense. Meta-SNP takes in a protein sequence as the input and predicts whether a mutation is disease-causing or neutral. Meta-SNP integrates the predictive capabilities of four distinct algorithms, namely PANTHER, PhD-SNP, SIFT, and SNAP [11]. On the other hand, Alphamissense relies on a deep learning model for its predictive power [12]. Alphamissense was developed by a scientist at Google DeepMind, and it leverages the AlphaFold algorithm to classify mutations as either pathogenic or benign. To assess any potential correlations between our Foldx outputs and the predictions from Meta-SNP and Alphamissense, we performed a comparative analysis. For statistical analysis and graph plotting, we utilized the RGui platform, specifically version 4.3.2.

## 3. Results

### 3.1. Phylogenetic Analysis of Gpx7 and GPx8

The Percentage Identity Matrix (PIM) in Figure 1A shows that the sequences of GPx7 and GPx8 are 50.27% identical. This is also evident in the phylogenetic tree in Figure 1B, where GPx7 and GPx8 emerge from a common GPx4 ancestor. Furthermore, GPx7 shows an earlier evolution than GPx8. Scientists believed that GPx7 and GPx8 evolved from GPx4 before the separation of mammals and fish. As shown in Figure 1C, the active sites (red rectangle) and the Glutathione Peroxidase Signature_2 domains (green rectangle) are perfectly conserved in GPx7 and GPx8. The active sites are 57C and 79C in GPx7 and GPx8, respectively. The Glutathione Peroxidase Signature_2 domains are 82^LAFPCNQF^89 and 104^LAFPCNQF^111 in GPx7 and GPx8, respectively. The region of the sequence in Figure 1C represents a high sequence similarity, residue by residue. Figure 1D shows a high structural similarity between GPx7 and GPx8 with a root mean square deviation (RSMD) of 0.528.

### 3.2. Classification and Distribution of Missense Mutations

Our analysis of missense mutations in the GPx7 protein revealed that approximately 70% of the 2926 mutations resulted in protein destabilization, as indicated by a ΔΔG value greater than 0.5. Conversely, around 7% of the mutations were found to stabilize the GPx7 protein structure, with a ΔΔG value lower than −0.5 (Figure 2). The distribution of energy change values, as depicted in the histogram, exhibited a median of 1.37 kcal/mol. Moreover, the mean energy change value for these mutations was calculated to be 2.53 kcal/mol. Regarding GPx8, our analysis showed that approximately 63% of the 3971 missense mutations led to protein destabilization, while approximately 8% had a stabilizing effect by lowering the Gibb’s free energy (Figure 2). The median energy change (ΔΔG) value for GPx8 was found to be 1.01 kcal/mol, with a mean value of 2.06 kcal/mol.

### 3.3. Computational Analysis of the Effect of Missense Mutations on GPx7 and GPx8

Figure 3A,B show the domains and sites, positional heatmaps, and line charts of GPx7 and GPx8, respectively. As indicated by the red arrows in Figure 3A, we observed three potential domains and one identified domain (Glutathione Peroxidase Signature_2) on the GPx7 heatmap. These four domains are located within the GSH_peroxidase domain and missense mutations in these regions greatly destabilize GPx7. The line chart highlights four critical residues: G153, A54, N80, and G58. The G153 and A54 residues are the most critical residues as any missense mutation would cause the greatest instability of GPx7. In contrast, missense mutations affecting the N80 and A54 residues would improve the stability of the GPx7 structure. Figure 3C shows the heatmap of the ten most critical residues and the energy values (ΔΔG) of each missense mutation within those residues. The missense mutations G153H and G153F increased the ΔG of the wild-type GPx7 structure by 49.53 kcal/mol and 39.39 kcal/mol, respectively.

The domain layout of GPx8 in Figure 3B is very similar to GPx7’s domain in Figure 3A. Within the GSH_peroxidase domain in GPx8, we highlighted, with red arrows, two unidentified domains and one identified domain (Glutathione Peroxidase Signature_2). Four critical residues of GPx8 stood out and were highlighted in the line chart: G175, P162, S157, and S99. Missense mutations in G175 and P162 tend to destabilize GPx8’s structure. In contrast, missense mutations in S157 and S99 stabilize GPx8. The heatmap in Figure 3D shows that N74W and G175W have the greatest destabilization impact by increasing the wild-type energy values by 41.12 kcal/mol and 35.87 kcal/mol, respectively. The top critical residues, G153 and G175, align, respectively, with the sequences of GPx7 and GPx8. Furthermore, the second most-critical residue on GPx8, P162, is also conserved and it aligns with residue P140 on GPx7. The A54 residue on GPx7 is conserved and aligns with A76 on GPx8.

The four conserved regions represent the beta-sheets in the GPx7 structure. The G153 residue is located at the downstream loop of a beta-sheet and its α-carbon is 5.4 Å away from the α-carbon of A150 residue. The A54 residue is close to F144 and F89 structurally. The distance between the α-carbons of A54 and F144 is 7.3 Å, and the α-carbon of A54 is 7.0 Å from F89 (Figure 3E). G175 and P162 are in the loop of the GPx8 structure. Like G153 on GPx7, G175 is on the upstream loop of the beta-sheet on the C-terminal of the GPx8. G175 connects with V171 across the loop, and they are 5.5 Å apart. P162 is in proximity with W164 (Figure 3F).

In our analysis, we conducted a comparison of the top missense mutations using the Meta-SNP tool. Additionally, we employed Foldx predictions to assess the impact of these mutations. The results indicate that destabilizing mutations are predicted to be pathogenic according to Meta-SNP (Table 1). However, upon further examination, we observed slight discrepancies in the Meta-SNP predictions for the top-five stabilizing mutations in both GPx7 and GPx8.

### 3.4. Somatic Mutations, Energy Changes, and Cancer Types

In our analysis, we observed a range of folding energy changes caused by missense mutations in GPx7-associated cancers. Figure 4 displays annotated missense mutations, along with the corresponding cancer types and their impact on folding energy. Interestingly, we noted distinct energy changes, even among mutations occurring in the same type of amino acids within similar cancer types. For instance, in small-intestine carcinoma, the G137V mutation was found to highly destabilize GPx7 with a folding energy change of 5.07 kcal/mol. Conversely, the G124S mutation had a nearly neutral effect with a minimal energy difference of 0.01 kcal/mol. These observations highlight the diverse impact of somatic mutations on the stability of GPx7, even within the same cancer type. In Figure 4C, we show that G137, C57, V168, and V176 are surface residues, and mutations here affect how GPx7 interacts with the environment or other proteins. Specifically, C57 is identified as an active site, and its substitution to tyrosine, known as C57Y, has been linked to skin adnexal tumor development as reported in the COSMIC database. This mutation, C57Y, has been shown to destabilize the GPx7 structure with a calculated ΔΔG of 2.38 kcal/mol. As shown in Figure 4D, GPx8 destabilizing somatic mutations, L104W and N172H, are located in the core of the protein while the stabilizing somatic mutations, S157F and D208N, are on the surface of the GPx8 protein structure.

Furthermore, our analysis revealed that most somatic mutations of GPx8 destabilize its structure. The L104W mutation was reportedly found in patients with carcinoma of the kidney, and our analysis showed that it caused the highest energy change of 24.03 kcal/mol. The GPx8 somatic mutation with the least energy change is S157F (ΔΔG = −1.23 kcal/mol) and was discovered in patients with skin carcinoma [13].

### 3.5. Comparative and Statistical Analyses

The boxplots in Figure 5 display the distribution of the Alphamissense and Meta-SNP scores within each category, allowing for an easy comparison of the central tendency, variability, and potential outliers. The highly destabilizing category (>2.5) for GPx7 and GPx8 shows the least variability when predicted by Meta-SNP and Alphamissense tools. The Analysis of Variance (ANOVA) yielded a *p*-value of 2 × 10^−16^ that provides strong evidence against the null hypothesis, indicating that there are significant differences in the Meta-SNP and Alphamissense scores across the five energy change categories.

## 4. Discussion

This scientific investigation provides valuable insights into the effects of specific rare missense mutations on the structural stability and functional implications of GPx7 and GPx8. While the experimental approach remains slow, tedious, and labor-intensive, we provided an alternative computational approach to investigate the role of missense mutations. This approach has been applied to several other proteins, including Myeloperoxidase [14] and Thyroid Peroxidase [15]. Most rare missense mutations have a very low frequency (<0.5%). We generated all possible missense mutations of GPx7 and GPx8, and some mutations have been identified in the literature while most remain unidentified. When compared with reliable and robust tools, our prediction shows a significantly high correlation with a *p*-value < 2 × 10^−16^. In this study, we highlighted ten missense mutations in Figure 3A,B with significant impacts on the structures of GPx7 and GPx8. These mutations have neither been investigated nor discovered. G175 and G153 are conserved residues in GPx8 and GPx7, respectively. We inferred that they are in an important domain that serves critical functions. The A54T mutation in GPx7 identified through genomic screening was associated with patients exhibiting intellectual disability [16], with a significant destabilizing effect (3.38 kcal/mol) on the protein. We also identified another mutation, A54W, in the same position with significantly high destabilizing features. The K182R mutations on GPx8 have been reported as an inheritable/Mendelian trait and as a somatic mutation. K182R as a germline variant increased the risk of epithelial ovarian cancer [17]. Moreover, K182R has been identified as a sporadic or somatic mutation associated with soft tissue tumor in the gastrointestinal tract [18]. K182R inactivates the mTORC1 repressor gene, DEPDC5, thereby driving the progression of duodenal cancer [19]. We predicted some somatic mutations to have a neutral effect on the stability of GPx7 and GPx8. However, it was shown that they are part of polygenic variants driving the disease phenotype. These findings establish a foundational framework to explore clinically relevant mutations, distinguishing those with potential driver roles from incidental, passenger mutations.

Based on our results, we demonstrate that most somatic mutations of GPx7 occur on surface residues, as in the cases of G137, C57, V168, and V176. For C57, it has been established that enzymes that utilize cysteine as an active site rely on its deprotonation for activity. Any alterations in cysteine at an active site may reduce the protein stability and inhibit the enzymatic function of the protein [20]. However, the role of these residues at the surface needs further investigation. The top destabilizing somatic mutation of GPx8 affects residues in highly conserved domains. The L104W mutation occurs on residue that is part of the Glutathione Peroxidase Signature_2 domain, and N172H interacts with V171 in the 164^WNFWKYLV^171 domain. This suggests a potential role of these mutations in compromising the stability and function of GPx8, potentially contributing to the development and progression of associated cancers. From the analysis of the heatmap, specific regions have been identified and designated as “unidentified domains”, which are deemed crucial for the functioning of GPx7 and GPx8. Structurally, these domains are integral components of the beta-sheet structure and exhibit conservation in both GPx7 and GPx8. The positioning of residues, whether within the hydrophobic core or the hydrophilic periphery, and their interactions with neighboring residues are pivotal in determining the potential impact of mutations. Typically, residues within 8 Å are considered contact residues, signifying their physical interaction with one another [21].

The residue G153 occupies a central position within the GPx7 structure and interacts with residues located within a conserved domain, notably including the A150 residue. Furthermore, the substitution of glycine at position 153 with tryptophan has been observed to escalate Van der Waal steric clashes, thereby distorting the structure of GPx7. Another significant interaction involves A54, which interacts with two phenylalanine residues, F89 and F144, situated within distinct conserved domains. Notably, F89 is associated with the glutathione Peroxidase Signature_2 domain, while F144 is located within a domain that has been identified as critical to the structure and function of GPx7. It has been observed that the substitution of alanine, characterized by a short hydrophobic sidechain, with a longer, polar, or aromatic side chain results in constriction, leading to instability in the GPx7 structure. In the case of GPx8, the residue G175, which is conserved within its structure, plays a crucial role owing to its hydrophobic nature and central location, thereby significantly contributing to the stability of GPx8. Most substitutions of G175 have been predicted to result in destabilization. Additionally, P162 has been identified as the second most-critical residue for GPx8, as it is situated within an upstream loop of a beta-sheet and interacts with numerous residues, including the conserved W164 residue. These findings underscore the paramount importance of the location and interactions of residues in comprehending the implications of mutations on protein structures, particularly within the context of GPx7 and GPx8.

Our statistical analysis, including ANOVA, reveals that the predicted stability effects of mutations are consistent with pathogenicity scores provided by AlphaMissense and Meta-SNP, tools chosen for their robust performance in similar research contexts. The ANOVA results highlight that highly destabilizing mutations are statistically more likely to be pathogenic or disease-causing compared to mutations with neutral or stabilizing effects. This correlation supports the robustness of our predictions, underscoring the potential of stability changes as a marker for pathogenicity. Further exploration of these associations could provide valuable insights into the molecular mechanisms underlying mutation-driven complex diseases. While these findings are promising, we recognize inherent limitations and potential biases in our computational approach. It is important to note that experimental studies of GPx7 and GPx8 structures would be invaluable, as they could reveal essential data and structural details that in silico methods may overlook. Additionally, it is crucial to recognize that mutations in other genes might also play a role in the development of cancers linked to GPx7 and GPx8, underscoring the complexity of the genetic factors involved. Future studies may incorporate additional predictive tools and experimental validation to further enhance the reliability of these predictions.

## 5. Conclusions

In conclusion, our comprehensive analysis of somatic mutations in GPx7 and GPx8 provides valuable insights into their impact on protein structure and stability. By combining computational methods and experimental data, we can further our understanding of the potential implications of these mutations in cancer development and progression. Understanding the molecular underpinnings of diseases, particularly at the level of somatic mutations, holds significant clinical relevance. By elucidating the structural and stability consequences of missense mutations, we will further study the biological significance of these mutation to uncover the pathogenic mechanisms involved in different cancer types. This knowledge can inform clinical diagnosis, treatment strategies, and aid in the development of targeted therapies.

## Figures and Tables

**Figure 1 cancers-17-00105-f001:**
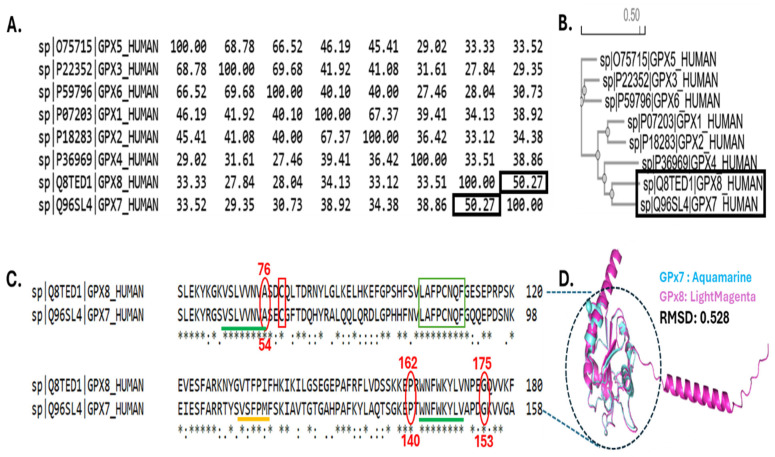
Evolutionary relationship between GPx7 and GPx8. (**A**) Percent Identity Matrix (PIM) of eight GPx proteins. The sequence identity between GPx7 and GPx8 is marked in black boxes. (**B**) Phylogenetic tree highlighting GPx7 and GPx8. (**C**) Pairwise alignment of GPx7 and GPx8. Green line and green box show conserved domains. The yellow line shows a critical domain for GPx7 only. Red oval shows conservation in critical residues for the stability of GPx7 and GPx8. (**D**) Structural alignment between GPx7 and GPx8. The dotted circle indicates a highly conserved region in GPx7 and GPx8.

**Figure 2 cancers-17-00105-f002:**
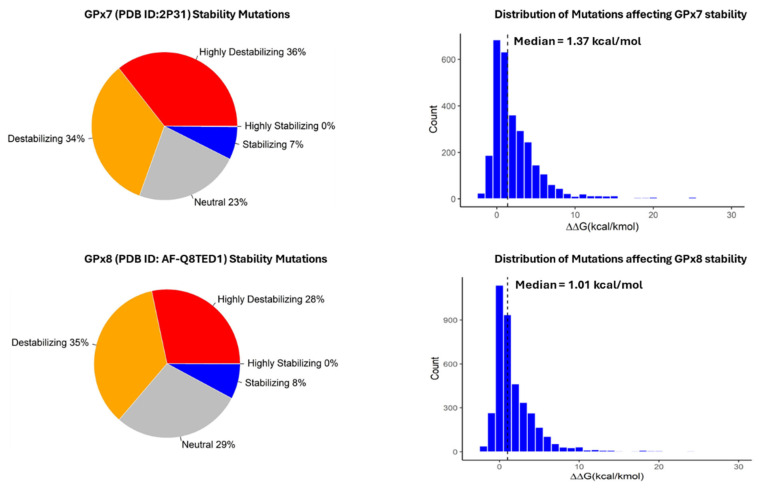
Distribution the effect of missense mutations on GPx7 (**top row**) and GPx8 (**bottom row**). The dotted lines on the histograms represent the median energy change.

**Figure 3 cancers-17-00105-f003:**
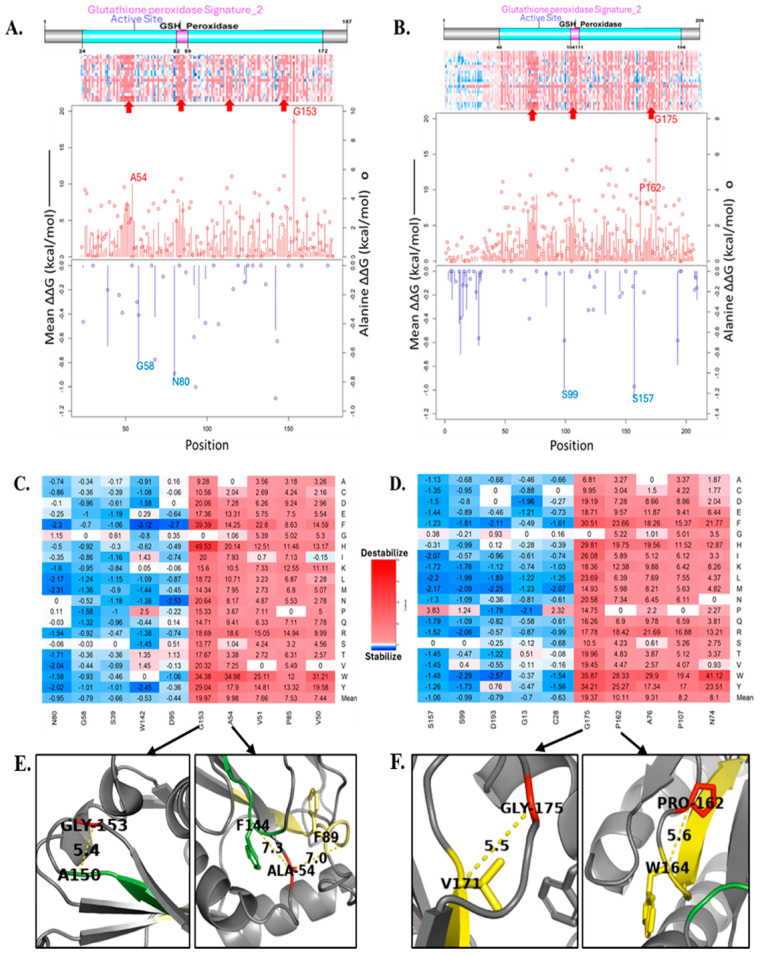
Gibb’s free energy change values and mutagenesis. Domain, complete heatmap, and line chart of (**A**) Gpx7 and (**B**) GPx8. Line represents mean ΔΔG values and the bubbles represent alanine mutagenesis. Red represents destabilizing missense mutations and blue represents stabilizing missense mutations. Top residues with missense mutations and their corresponding ΔΔG values for (**C**) GPx7 and (**D**) GPx8. (**E**) GPx7 structure and (**F**) Gpx8 structure showing intermolecular distances in Angstrom between critical residues in red and neighboring residues in yellow.

**Figure 4 cancers-17-00105-f004:**
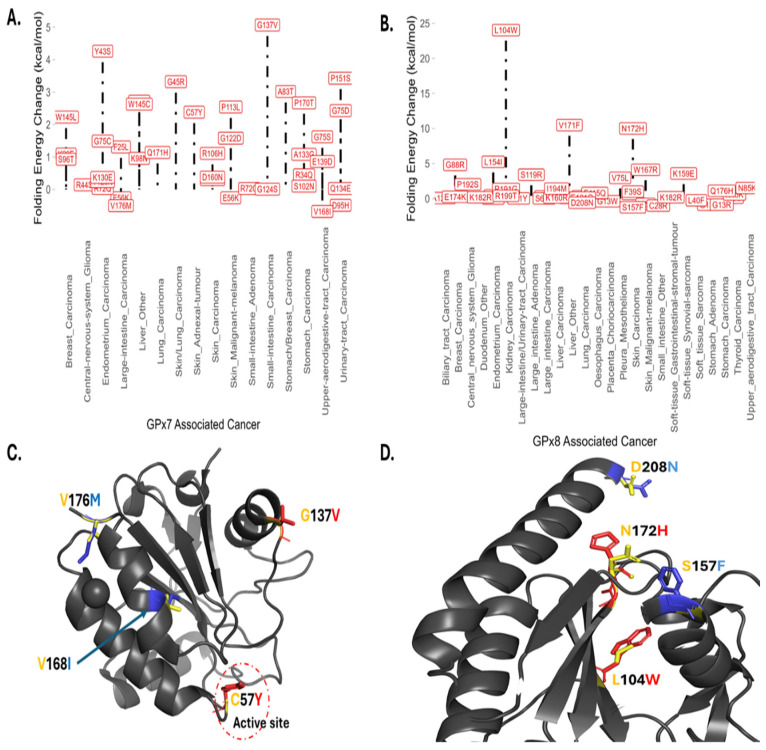
Somatic mutations. Lollipop plot of Foldx predictions of cancer-causing somatic mutations in (**A**) GPx7 and (**B**) GPx8. Structural localization of top destabilizing (red) and top stabilizing (blue) somatic mutations in (**C**) GPx7 and (**D**) GPx8. Wild-type residues in yellow.

**Figure 5 cancers-17-00105-f005:**
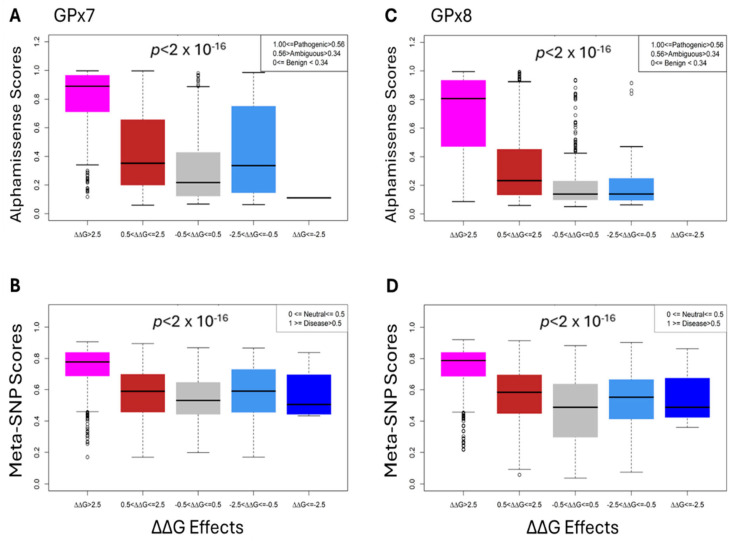
Statistical analysis of our Foldx predictions for Alphamissense (top row (**A**,**C**)) and Meta-SNP (bottom row (**B**,**D**)).

**Table 1 cancers-17-00105-t001:** Comparative prediction of top-five destabilizing and stabilizing missense mutations.

Top 5 Destabilizing Mutations					Top 5 Stabilizing Mutations				
GPx8	Foldx		Meta-SNP		GPx8	Foldx		Meta-SNP	
Mutations	∆∆G	Effect	Score	Pathogenicity	Mutations	∆∆G	Effect	Score	Pathogenicity
N74W	41.12	Highly_Des	0.89	Disease	S70I	−2.82	Highly_Sta	0.36	Neutral
G175W	35.87	Highly_Des	0.88	Disease	S119P	−2.74	Highly_Sta	0.49	Neutral
G175Y	34.21	Highly_Des	0.88	Disease	D193W	−2.57	Highly_Sta	0.86	Disease
A126W	33.09	Highly_Des	0.82	Disease	P118D	−2.39	Stabilizing	0.30	Neutral
G175F	30.51	Highly_Des	0.87	Disease	S99W	−2.29	Stabilizing	0.72	Disease
**GPx7**					**GPx7**				
**Mutations**	**∆∆G**	**Effect**	**Score**	**Pathogenicity**	**Mutations**	**∆∆G**	**Effect**	**Score**	**Pathogenicity**
G153H	49.53	Highly_Des	0.89	Disease	E93M	−3.61	Highly_Sta	0.56	Disease
G153F	39.39	Highly_Des	0.87	Disease	W142F	−3.12	Highly_Sta	0.84	Disease
A54W	34.98	Highly_Des	0.88	Disease	D95F	−2.70	Highly_Sta	0.45	Neutral
G153W	34.38	Highly_Des	0.89	Disease	D95N	−2.53	Highly_Sta	0.43	Neutral
V50W	31.21	Highly_Des	0.86	Disease	S48I	−2.49	Stabilizing	0.40	Neutral

## Data Availability

The data supporting this study are available within the paper. Other data are available from the authors upon request.

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
