# Peer review of "Computational Mutagenesis of GPx7 and GPx8: Structural and Stability Insights into Rare Genetic and Somatic Missense Mutations and Their Implications for Cancer Development"

_cancers, 2024, doi:10.3390/cancers17010105_

Round 1

Reviewer 1 Report

Comments and Suggestions for Authors

Glutathion Perioxidases, protein structure and stability, protein folding in endoplasmatic is of crucial biological importance, and energy consideration is very fundamental for the understanding of biology, and the submitted publication gives important contribution. 

Author Response

Glutathion Perioxidases, protein structure and stability, protein folding in endoplasmatic is of crucial biological importance, and energy consideration is very fundamental for the understanding of biology, and the submitted publication gives important contribution. 

>>We sincerely thank the reviewer for the positive feedback and recognition of the importance of our findings.

Reviewer 2 Report

Comments and Suggestions for Authors

The authors have presented a well-thought out strategy to conduct computational mutagenesis to analyse the mutation status of GPx7 and GPx8 to determine which mutations are most pathogenic.

The "Introduction" provided sufficient information on the biology of GPx7 and GPx8 with appropriate references. The methods section was reproducible and provided excellent instruction, particularly someone who wants to use this for their gene of interest.

The presentation of scientific argument was high with critical evaluation, particularly the information in Figure 4. 

I agree that "this scientific investigation provides valuable insights into the effects of specific rare 284 missense mutations on the structural stability and functional implications of GPx7 and 285 GPx8."  

The manuscript is well written with an excellent study design.  I note that they have used the same published methodology but two different proteins. I"m assuming that they are using own citations as no one else has published a similar strategy.   

I support the publication of this manuscript. It also fits the criteria for the special edition.  I thought the science was very strong, the figures were easy to interpret and the data was very informative and helpful to those who are curating mutations.  

Minor Corrections:

Figure 3A and B: Increase font size for headings blue and pink writing above the bar

Author Response

The authors have presented a well-thought out strategy to conduct computational mutagenesis to analyse the mutation status of GPx7 and GPx8 to determine which mutations are most pathogenic.

The "Introduction" provided sufficient information on the biology of GPx7 and GPx8 with appropriate references. The methods section was reproducible and provided excellent instruction, particularly someone who wants to use this for their gene of interest.

The presentation of scientific argument was high with critical evaluation, particularly the information in Figure 4. 

I agree that "this scientific investigation provides valuable insights into the effects of specific rare 284 missense mutations on the structural stability and functional implications of GPx7 and 285 GPx8."  

The manuscript is well written with an excellent study design.  I note that they have used the same published methodology but two different proteins. I"m assuming that they are using own citations as no one else has published a similar strategy.   

I support the publication of this manuscript. It also fits the criteria for the special edition.  I thought the science was very strong, the figures were easy to interpret and the data was very informative and helpful to those who are curating mutations.  

Minor Corrections:

Figure 3A and B: Increase font size for headings blue and pink writing above the bar

>>We sincerely thank the reviewer for their positive feedback and thoughtful evaluation of our manuscript. Regarding the minor correction, we have increased the font size for the headings in blue and pink above the bars in Figures 3. We thank the reviewer for bringing this to our attention.

Reviewer 3 Report

Comments and Suggestions for Authors

This manuscript by Sobitan and colleagues examines the effects of mutations on the structures of GPx7 and GPx8 using computational methods. They have essentially repeated studies already published, using similar methods, of Myeloperoxidase and Thyroid Peroxidase. As the authors state physical experimental protocols are slow, laborious and time consuming; however my personal opinion is that in silico methods are not always an equivalent replacement. 

However, as far as it goes this is a satisfactory study which only requires a very minor modification. The authors should make the point at the beginning of the Discussion that actual physical studies of the proteins' structures would be very useful and might produce data not seen in the computational study. Also the point should be made that mutations in other genes could have contributed to the cancers described (with mutations in GPx7 and GPx8).

Author Response

This manuscript by Sobitan and colleagues examines the effects of mutations on the structures of GPx7 and GPx8 using computational methods. They have essentially repeated studies already published, using similar methods, of Myeloperoxidase and Thyroid Peroxidase. As the authors state physical experimental protocols are slow, laborious and time consuming; however my personal opinion is that in silico methods are not always an equivalent replacement. 

However, as far as it goes this is a satisfactory study which only requires a very minor modification. The authors should make the point at the beginning of the Discussion that actual physical studies of the proteins' structures would be very useful and might produce data not seen in the computational study. Also the point should be made that mutations in other genes could have contributed to the cancers described (with mutations in GPx7 and GPx8).

>>We appreciate the reviewer’s thoughtful feedback. To address the reviewer’s concerns and further strengthen the manuscript, we have added the following sentences to the Discussion section:

“It is important to note that experimental studies of GPx7 and GPx8 structures would be invaluable, as they could reveal essential data and structural details that in silico methods may overlook. Additionally, it is crucial to recognize that mutations in other genes might also play a role in the development of cancers linked to GPx7 and GPx8, underscoring the complexity of the genetic factors involved.”